# Risk factors for early suspected ventilator-associated pneumonia in severe thoracic blunt trauma patient: A French national cohort study

Gary Duclos[1]*, Victor Vidal[1], Noemie Resseguier[2,3], Mathieu Boutonnet[4], Thomas Botrel[5], Gerard Audibert[6], Manon Séguret[7], Vincent Legros[8], Caroline Jeantrelle[9], Anh Dao Phan[10], Olivier Langeron[11], Elisabeth Gaertner[12], Jean-Luc Hanouz[13], Thomas Clavier[14], Véronique Ramonda[15], Fanny Bounes[16], Paër-selim Abback[17], Mathieu Willig[18], Julien Pottecher[19], Marc Leone[20], for Traumabase® Group

1 Service d'anesthésie et de réanimation traumatologique, Assistance – Publique Hôpitaux de Marseille, Hôpital Nord, France Aix- Marseille Université, Marseille, France, 2 Service d'Épidémiologie et d'Économie de la Santé, AP-HM, France Aix- Marseille Université, Marseille, France, 3 CEReSS-Health Services and Quality of Research, Aix-Marseille University, Marseille, France, 4 Service de réanimation, Hôpital d'Instruction des Armées Percy, Service de santé des Armées, Clamart, France, 5 Anaesthesia and Intensive Care Department, AP-HP, Pitié-Salpêtrière Hospital, Paris, France, 6 Department of Anesthesiology and Surgical Intensive Care, CHRU-Nancy, Université de Lorraine, Nancy, France, 7 Service d'Anesthésie-Réanimation Chirurgicale, Hôpital Bicêtre, Université Paris-Saclay, Assistance Publique - Hôpitaux de Paris, Le Kremlin-Bicêtre, France, 8 Department of Anesthesiology and Critical Care, Reims University Hospital, Hôpital Robert Debré - CHU de Reims, Rue Koenig, Reims, France, 9 Department of Anesthesiology and Critical Care, Beaujon Hospital, DMU Parabol, AP-HP.Nord, Paris, France, 10 Department of Anesthesiology and Critical Care Medicine, Hôpital Européen Georges Pompidou, Assistance Publique-Hôpitaux de Paris, Paris, France, 11 Department of Anesthesia and Intensive Care, Henri Mondor University Hospital, Assistance Publique - Hôpitaux de Paris (APHP), University Paris-Est Créteil (UPEC), Paris, France, 12 Hôpital Louis Pasteur, Service d'Anesthésie-Réanimation Pôle 2, 39 Avenue de la Liberté, Colmar, France, 13 Department of Anesthesiology and Critical Care Medicine, Caen University Hospital, Avenue de la cote de Nacre, Caen, France, 14 Department of Anesthesiology, Critical Care and Perioperative Medicine, Surgical Intensive Care Unit, Rouen University Hospital, Rouen, France, 15 Department of Anesthesiology and Critical Care, University Toulouse 3-Paul-Sabatier, University Hospital of Toulouse, Hôpital Pierre-Paul Riquet, CHU Toulouse-Purpan, Toulouse, France, 16 Anesthesiology and Intensive Care Department CHU Toulouse, Toulouse, France, 17 Department of Anesthesiology and Critical Care Medicine, CHU Tours, Tours University Hospital, Tours, France, 18 Anaesthesiology and Critical Care Department, Dijon Bourgogne University Hospital, Dijon, France, 19 Department of Anaesthesiology, Critical Care and Perioperative Medicine, Fédération de Médecine Translationnelle de Strasbourg, ER, Strasbourg University Hospital, Strasbourg, France, 20 Service d'anesthésie et de réanimation traumatologique, Assistance – Publique Hôpitaux de Marseille, Hôpital Nord, Marseille, France Aix- Marseille Université, Marseille, France

* Gary.duclos@ap-hm.fr

## Abstract

### Background

Ventilator-associated pneumonia (VAP) is the most common infection in severely injured patients requiring mechanical ventilation. Chest trauma has been identified as a significant risk factor for VAP. This study aimed to describe the risk factors for early VAP in patients with severe blunt thoracic trauma admitted to the intensive care unit (ICU) and receiving mechanical ventilation.

**Data availability statement:** Data cannot be shared because of french legislation . Data are available from the Comité informatique et liberté Institutional Data Access for researchers who meet the criteria for access to confidential data on reasonable request to contact@traumabase.eu

**Funding:** The author(s) received no specific funding for this work.

**Competing interests:** The authors have declared that no competing interests exist.

## Materials and Methods

A retrospective cohort study was conducted using data from a national registry including data from 17 French trauma centers during a period of seven years. The study included patients with severe blunt thoracic trauma requiring invasive mechanical ventilation. Data analysis focused on identifying independent risk factors for early suspected VAP (occurring within 48 hours to 5 days after ICU admission) using two models of logistic regression.

## Results

From 31700 patients screened 712 patients were analyzed. Early suspected VAP occurred in 192 (27%) patients. The study identified several independent risk factors associated with early suspected VAP in patients with severe blunt thoracic trauma: male gender (OR= 2.77, 95%CI: 1.68–4.77, $p < 0.001$), ASA score >1 (OR= 1.64, 95%CI: 1.08–2.50, $p = 0.019$), injury severity score (ISS) >15 (OR=3.15, 95%CI: 1.13–11.99, $p = 0.025$), initial Glasgow Coma Scale (GCS) score <9 (OR=2.71, 95%CI: 1.88–3.96, $p < 0.001$), absolute thoracic abbreviated injury scale (AIS) (OR=1.51, 95%CI: 1.14–1.99, $p = 0.003$), and the number of packed red blood cells (PRBCs) transfused within the first 24 hours (OR=1.04, 95%CI: 1.00–1.08, $p = 0.027$). Prehospital antibiotic administration was identified as a protective factor (OR=0.54, 95%CI: 0.29–0.94, $p = 0.028$).

## Conclusion

In patients with severe blunt chest trauma receiving invasive mechanical ventilation, male gender, ASA score, ISS > 15, GCS < 9, thoracic AIS and number of PRBCs transfused were independent risk factors for early suspected VAP. Prehospital antibiotic therapy was a protective factor, suggesting potential strategies for VAP prevention.

## Background

Ventilator-associated pneumonia (VAP) is the commonest healthcare-associated infections in severely injured patients receiving mechanical ventilation [1]. In 2020, a retrospective study including a heterogeneous population of 1403 severely injured patients highlighted that the risk for VAP increased with age, the need for massive transfusion and the presence of facial, spinal, and sternal injuries [2]. A multicenter study identified chest injuries as an independent risk factor for VAP in patients with severe traumatic brain injuries [3]. To date, only one Dutch large retrospective study assessed the risk of pneumonia among 1162 patients with severe thoracic trauma (defined by an abbreviated injury scale (AIS) of thoracic region of 2 or more). [4] The incidence of pneumonia was 27.5% and the risk factors were age, male gender, and duration of mechanical ventilation, while severity criteria scores and comorbidities

were not associated with this risk [4]. However, this study included heterogeneous patients: patients with and without tracheal intubation and those with blunt and penetrating injuries were indifferently included. Moreover, confounding factors such as coma or admission under invasive mechanical ventilation were not considered [4].

Our study aimed to describe risk factors for suspected VAP occurring within the first five days after intensive care unit (ICU) admission in a homogeneous, highly characterized population of patients with severe blunt thoracic trauma requiring mechanical ventilation.

## Materials and methods

### Study design

We conducted a retrospective cohort study using prospectively collected data recorded in the French trauma registry Traumabase® [5,6]. The registry has obtained authorization from the Comité national informatique et libertée (CNIL) and is registered under number 911461. This study was approved by a Research Ethics Committee and is registered under the IRB number 00006477. We included patients from 17 centers contributing to the Traumabase® and agreeing to participate. The study period covers from January 2014 to December 2021.The conduct of this study followed the STrengthening the Reporting of OBservational studies (STROBE) recommendations [7].

### Inclusion and exclusion criteria

Inclusion occurred with three criteria present: 1) Primary admission to trauma resuscitation room, 2) blunt thoracic trauma defined by an AIS > 2 and 3) early invasive mechanical ventilation initiated in the prehospital setting or trauma resuscitation room for at least five days.

Excluded patients were those under 18 years old, with any kind of penetrating trauma and those who had a cardiac arrest before hospital admission. To minimize the exposure bias, the patients who died within the first 5 days after ICU admission were also excluded. Suspected pneumonia declare before day 2 and after day 5 were by instanced not included.

### Outcome

The diagnosis of suspected VAP was based on international guidelines [8–10]. It was suspected in case of systemic infection (fever (>38°C) or hypothermia (<36°C) or leukocytosis (≥12,000 cells/mm$^3$) or leukopenia (≤4000 cells/mm$^3$)), a worsening of gas exchange and a new onset or progressive infiltrate at the chest X-ray. To improve the consistency of this diagnosis, Physician investigators in each center were asked to declare the presence of an episode of pneumonia in the registry according to international criteria and antibiotic treatment. As no microbiological documentation was available in the data base we have chosen to qualify the diagnosis of VAP as suspected.

Early suspected VAP was considered if the patient developed pneumonia after 48 hours of initiation of mechanical ventilation and before the end of the 5th day after ICU admission [10]. In case of missing data on this information, the patient was excluded from the analysis.

### Definitions of analyzed variables

The following quantitative variables were considered for analysis: age, weight, height, body mass index (BMI), American Society of Anesthesiologists (ASA) score, initial Glasgow coma scale (GCS) score, and initial motor GCS score. Prehospital data were vital signs on scene (systolic and diastolic blood pressure, heart rate, transcutaneous oxygen saturation), and their worst values during transport, initial point-of-care hemoglobin, volumes of infused crystalloids, administered blood products, and maximal flow of catecholamine infused. Hospital data included vital signs at the trauma bay admission (systolic and diastolic blood pressure, heart rate, transcutaneous oxygen saturation, respiratory rate,

expired carbon dioxide ($CO_2$), body temperature), laboratory values, pH, arterial $PaO_2$ and $PaCO_2$, lactatemia, hemoglobin, platelet, prothrombin time, fibrinogen, creatinine, bicarbonate, troponin, alcohol), simplified acute physiology score (SAPS), sequential organ failure assessment (SOFA) score and detail of each of its components, lowest $PaO_2$:$FiO_2$ ratio, number of intubation, duration of mechanical ventilation, day of first pneumonia, overall and critical care length of hospital stay (in days). The intra-hospital mortality rate was assessed at day 30.

We collected pre-hospital parameters and treatments including use of catecholamine, decompression of tension pneumothorax, transfusion and type of transfused blood product in the first 24 hours (including the number of packed red blood cells (PRBCs), fresh frozen plasma (FFP) and concentrate of platelets (CP)) and use of pre-hospital antibiotic therapy without precision.

The following variables were also collected in the first 24 hours of admission in the trauma bay: chest drainage, transfusion, massive transfusion defined by transfusion of more than 10 PRBCs units in the first 24 hours, surgery within the first 24 hours.

During the ICU stay, the following variables were collected: ICU mortality, withdrawal or withholding of life-sustaining therapy decision, use of invasive mechanical ventilation, use of tracheostomy, acute respiratory distress syndrome (ARDS), sepsis, septic shock, thoracic or abdominal surgery, and date of the first pneumonia.

For each patient, an injury severity score (ISS) and AIS code of traumatic injury was computed by each center [11,12]. For the need of this study, the ISS codes corresponding to hemothorax, unilateral or bilateral lung contusions, and rib flail were extracted and analyzed.

## Statistical Analysis

Statistical analysis was conducted with R version 4.0.5 (R Core Team (2021). R: A language and environment for statistical computing. R Foundation for Statistical Computing, Vienna, Austria). Two groups were defined: a group with early suspected VAP and a group without early suspected VAP. Quantitative variables were described as means and standard deviations (SD) or medians and 1st-3rd quartiles depending on their distribution. Univariate analyses were conducted to identify factors associated with the occurrence of early suspected VAP. Comparisons between the two groups were made using a Student's t-test when the conditions were met or a non-parametric Mann-Whitney test when they were not. Qualitative values were described as numbers (percentage) and compared between the two groups using a chi-square test for qualitative variables when the conditions were met or a non-parametric Fisher's test otherwise. A multivariate logistic regression model was established including potential confounding risk factors. Potential risk factors were selected a priori based on literature data. No selection based on statistical criteria was then performed. Adjusted odds ratios (OR) were estimated, along with their 95% confidence intervals (CI).

To increase the sensitivity of the analysis and avoid variable redundancy, two multivariate models were run to identify factors independently associated with the risk of early suspected VAP.

- The first model included variables such as age, male gender, ASA score >1, absolute thoracic AIS value (3, 4 or 5), abdominal AIS > 2, pre-hospital antibiotic administration, presence of spinal cord injury with neurological deficit, massive transfusion, and GCS score <9 at initial management.

- The second model included variables such as age, male gender, ASA score >1, ISS > 15, pre-hospital antibiotic administration, presence of spinal cord injury with neurological deficit, number of PRBCs transfused in the first 24 hours, GCS score <9 at initial management, presence of hemothorax, presence of a rib fail, and presence of lung contusion.

Multivariate regression models were checked for non-multicollinearity by estimating variance inflation factors and for goodness-of-fit by using Hosmer-Lemeshow test. Patients with missing data regarding one item included in the models were excluded of the analysis.

## Results

### Patient characteristics

During the study period, 31700 patients were registered in the Traumabase®. We screened 2037 chest trauma patients from 17 centers responding to the inclusion criteria. A cohort of 712 patients was included in our analysis after application of non-inclusion criteria (Fig 1: flow chart). The mean (SD) age was 39 (17) years, 569 (80%) patients were males, and the mean ISS was 31 (11). Two hundred fifty-one (35%) patients developed an infection during ICU stay, including 187 (26%) patients who developed early suspected VAP. The day-30 mortality was 8%. Main characteristics of patients included in the analysis are resumed in Table 1.

### Univariate analyses

The patients with early suspected VAP were more frequently males, had higher ISS, SOFA score, and SAPS II, and did not receive prehospital antibiotic therapy (Table 1). They received a longer duration of invasive mechanical ventilation and developed more commonly ARDS and septic shock (Table 1). No difference was found for mortality rate at day 30 between the patients with early suspected VAP and those with no early suspected VAP (Table 1).

### Multivariate analyses

Results from the two multivariable models are summarized in Table 2. Twenty-five patients were excluded because of missing data on at least one of the variables included in the models. In brief, male gender, ASA score and initial GCS score < 9 were independent factors associated with early suspected VAP in both models (Table 2). Conversely, prehospital

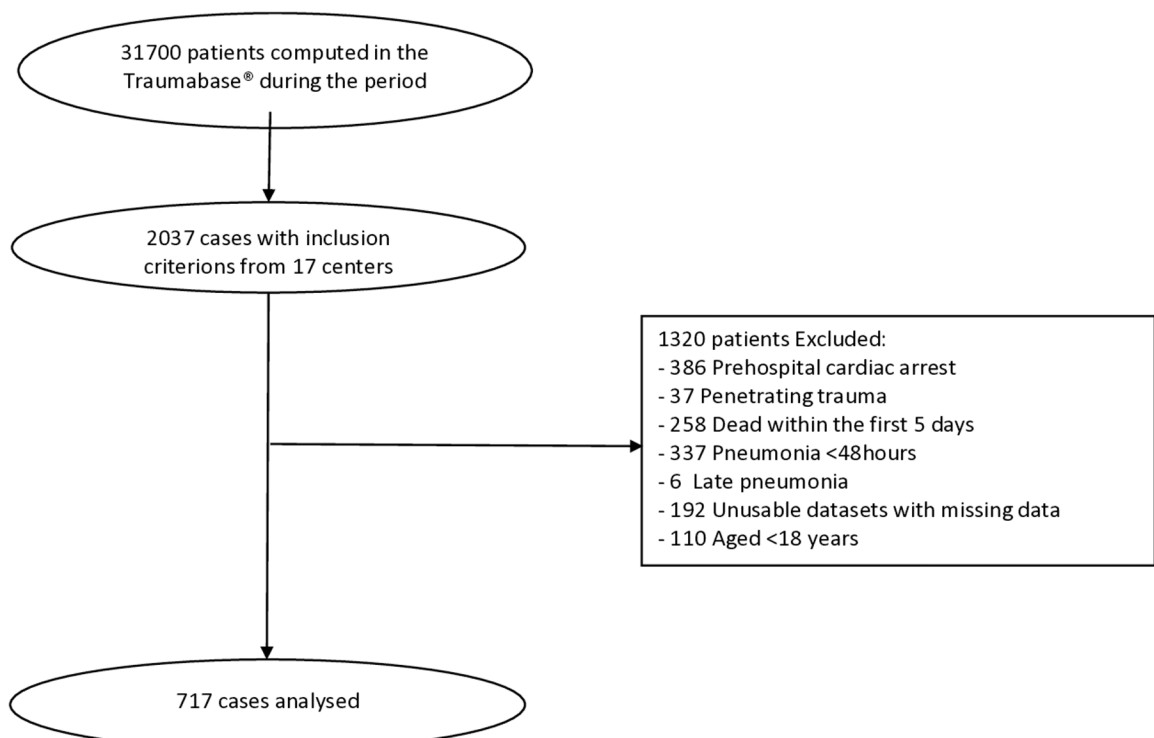

**Fig 1.** Flowchart of the study.

**Table 1. Descriptive statistics and univariate analyses comparing the early VAP and no early VAP groups.** Quantitative variables are expressed in medians and interquartile ranges (25-75) or in means ± standard deviations depending on their distribution.

| Variable | Sample size (n) | Total population (n = 712) | Suspected early VAP (n = 187) | Control group (n = 525) | p |
|---|---|---|---|---|---|
| Age (years) | 712 | 39 ± 17 | 39 ± 17 | 38 ± 17 | 0.402 |
| Male gender (%) | 711 | 569 (80%) | 166 (89%) | 403 (77%) | <0.001 |
| BMI value | 611 | 25 ± 4 | 25 ± 5 | 25 ± 4 | 0.314 |
| ASA score | 697 | 1.6 ± 1.2 | 1.7 ±1.2 | 1.6 ± 1.1 | 0.200 |
| ASA score >1 | 697 | 243 (35%) | 77 (42%) | 166 (32%) | 0.021 |
| Initial Glasgow scale <9 | 704 | 367 (52%) | 128 (68%) | 239 (46%) | <0.001 |
| Initial Glasgow scale value | 704 | 8.9 ± 4.4 | 7.3 ± 4.1 | 9.4 ±4.3 | <0.001 |
| Initial Glasgow motor scale value | 660 | 4.1 ± 2.0 | 3.4 ± 2.0 | 4.4 ±1.9 | <0.001 |
| Initial SBP (mmHg) | 670 | 122 ± 30 | 124 ± 33 | 122 ± 29 | 0.494 |
| Intial DBP (mmHg) | 669 | 74 ± 20 | 73 ± 21 | 74 ± 20 | 0.596 |
| Initial HR (mmHg) | 672 | 101 ± 28 | 104 ± 29 | 100 ± 27 | 0.119 |
| Initial SpO2 (mmHg) | 712 | 95 ± 6 | 93 ± 8 | 96 ± 5 | 0.003 |
| SOFA Respiratory value | 687 | 1.8 ± 1.3 | 2.4 ± 1.3 | 1.6 ± 1.3 | <0.001 |
| SOFA Coagulation value | 683 | 0.60 ± 0.84 | 0.68 ±0.89 | 0.57 ± 0.82 | 0.127 |
| SOFA Cardiovascular value | 686 | 3.0 ± 1.5 | 3.6 ± 1.1 | 2.8 ± 1.6 | <0.001 |
| SOFA Central nervous system value | 686 | 2.0 ± 1.5 | 2.5 ± 1.5 | 1.8 ± 1.5 | <0.001 |
| SOFA liver value | 666 | 0.23 ± 0.56 | 0.35 ± 0.66 | 0.18 ±0.51 | <0.001 |
| SOFA Renal function value | 683 | 0.30 ± 0.59 | 0.39 ± 0.63 | 0.27 ± 0.58 | 0.0193 |
| Total SOFA score | 709 | 8.3 ± 3.5 | 10.3 ± 3.0 | 7.6 ± 3.4 | <0.001 |
| ISS >15 (%) | 712 | 673 (95%) | 184 (98%) | 489 (93%) | 0.007 |
| ISS value | 712 | 31 ± 11 | 37 ± 11 | 28 ± 10 | <0.001 |
| AIS Head value | 712 | 2.6 ± 1.8 | 3.5 ±1.5 | 2.2 ± 1.7 | <0.001 |
| AIS Face value | 712 | 0.77 ± 1.05 | 0.87 ± 1.10 | 0.73 ± 1.03 | 0.107 |
| AIS Thorax value | 712 | 3.5 ± 0.6 | 3.6 ± 0.7 | 3.4 ± 0.6 | <0.001 |
| AIS Abdomen value | 712 | 1.2 ± 1.5 | 1.4 ± 1.5 | 1.2 ± 1.4 | 0.037 |
| AIS Abdomen >2 (%) | 712 | 146 (21%) | 49 (26%) | 97 (18%) | 0.025 |
| SAPSII Score | 709 | 45 ± 16 | 53 ± 14 | 42 ± 16 | <0.001 |
| PRBC unit transfused in 24h | 712 | 2.1 ± 5.0 | 2.9 ± 6.4 | 1.8 ± 4.4 | 0.011 |
| FFP units transfused in 24h | 712 | 1.6 ± 4.1 | 2.5 ± 5.3 | 1.3 ± 3.4 | <0.001 |
| Platelet unit transfused in 24h | 712 | 0.30 ± 1.41 | 0.49 ± 1.48 | 0.23 ± 1.39 | 0.030 |
| Massive transfusion (%) | 712 | 45 (6%) | 18 (10%) | 27 (5%) | 0.031 |
| Flail chest (%) | 712 | 74 (10%) | 26 (14%) | 48 (9%) | 0.067 |
| Pulmonary contusion (uni or bilateral) (%) | 712 | 184 (26%) | 58 (31%) | 126 (24%) | 0.060 |
| Hemothorax (%) | 712 | 179 (25%) | 59 (32%) | 120 (23%) | 0.019 |
| Spinal cord injury with ND (%) | 712 | 40 (6%) | 14 (7%) | 26 (5%) | 0.196 |
| Chest drainage (%) | 712 | 83 (12%) | 34 (18%) | 49 (9%) | 0.001 |
| Thoracic surgery (%) | 712 | 24 (3%) | 7 (4%) | 17 (3%) | 0.742 |
| Rib fixation surgery (%) | 712 | 8 (1%) | 3 (2%) | 5 (1%) | 0.439 |
| Abdominal surgery (%) | 164 | 26 (16%) | 4 (11%) | 22 (17%) | 0.340 |
| Prehospital antibiotic administration (%) | 712 | 91 (13%) | 16 (9%) | 75 (14%) | 0.044 |
| Sepsis during ICU stay (%) | 712 | 251 (35%) | 187 (100%) | 64 (12%) | <0.001 |
| Abdominal sepsis (%) | 712 | 29 (4%) | 22 (12%) | 7 (1%) | <0.001 |
| Urinary sepsis (%) | 712 | 27 (4%) | 14 (7%) | 13 (2%) | 0.002 |

*(Continued)*

**Table 1.** (Continued)

| Variable | Sample size (n) | Total population (n = 712) | Suspected early VAP (n = 187) | Control group (n = 525) | p |
|---|---|---|---|---|---|
| Neuro-meningeal sepsis (%) | 712 | 5 (<1%) | 2 (1%) | 3 (<1%) | 0.611 |
| Device-related sepsis (%) | 712 | 71 (10%) | 51 (27%) | 20 (4%) | <0.001 |
| Sepsis related to surgical site infection (%) | 712 | 50 (7%) | 16 (9%) | 34 (6%) | 0.339 |
| Septic shock during icu stay (%) | 712 | 60 (8%) | 49 (26%) | 11 (2%) | <0.001 |
| Days of mechanical ventilation | 685 | 6 [2 ; 19] | 26 [20 ; 38] | 3 [1 ; 7] | <0.001 |
| Tracheotomy (%) | 693 | 148 (21%) | 95 (51%) | 53 (10%) | <0.001 |
| ARDS (%) | 712 | 142 (20%) | 82 (44%) | 60 (11%) | <0.001 |
| ICU LOS (days) | 712 | 13 [6 ; 29] | 39 [27 ; 51] | 9 [5 ; 15] | <0.001 |
| Hospital LOS (days) | 658 | 28 [13 ; 48] | 50 [36 ; 71] | 19 [11 ; 36] | <0.001 |
| Death in critical care | 712 | 71 (10%) | 27 (14%) | 44 (8%) | 0.018 |
| 30-day mortality | 712 | 52 (8%) | 10 (5%) | 42 (8%) | 0.186 |

BMI: Body Mass Index; ASA: American society of anesthesiologists; SBP: Systolic blood pressure, DBP: diastolic blood pressure; SpO2: Pulse oximetry Oxygen saturation; SOFA: Sequential organ failure assessment score; ISS:Injury Severity score, AIS: Abbreviated Injury Scale; SAPSII: Simplified acute physiology score; PRBC 24h: number of packed red blood cells administered in the first 24hours; FFP 24h: number of fresh frozen plasma administered in the first 24hours; Platelet unit 24h: number of platelet unit administered in the first 24hours; Spinal cord injury with ND: Spinal cord injury with neurological deficit; ARDS: acute respiratory distress syndrome; ICU: intensive care unit; LOS: length of stay.

administration of antibiotic was the only protective factor (Table 2). The ISS, the absolute thoracic AIS value, the number of PRBCs transfused in the first 24 hours were significantly associated with early suspected VAP (Table 2). No significant association were found with age, rib fractures, hemothorax, contusion, abdominal AIS > 2, and spinal injury with neurologic deficit (Table 2). Variance inflation factors were low for each model (Table 2).

## Discussion

In our cohort, independent risk factors associated with early suspected VAP in patients with severe blunt thoracic trauma receiving invasive mechanical ventilation were male gender, ASA score >1, ISS > 15, absolute thoracic AIS value, initial GCS score < 9 and number of transfused PRBCs in the first 24 hours. Prehospital administration of antibiotic therapy was an independent protective factor. To our knowledge, this is the first study reporting risk factors for early suspected VAP in a homogeneous population of blunt thoracic trauma patients receiving invasive mechanical ventilation.

To date, Wutzler *et al.* in one of the largest multicentric retrospective cohort study (n = 1162) in this question, identified age, male gender, inhalation of gastric content, and duration of invasive mechanical ventilation as independent risk factors for pneumonia [4]. Unlike this study, we considered GCS score < 9 rather than inhalation. Indeed, a GCS score < 9 represents a risk factor for inhalation and this can be easily identified, unlike with reported pre-hospital inhalation [13–15]. In this study, severity scores such as ISS or thoracic AIS did not appear as independent risk factors in contrast to our study and others [2,15,16]. In fact, the most severe patients could have died before the onset of pneumonia, thus inducing a non-exposure bias. Furthermore, the heterogeneity of patients (blunt and penetrating trauma and patients with and without invasive mechanical ventilation) and the heterogeneity of the type of pneumonia probably explains this divergence.

Our findings showed that an initial GCS score < 9 was an independent risk factor associated with early suspected VAP. This association was previously described in the literature. In a single-center study of 221 severely thoracic trauma patients (ISS > 16 and thoracic AIS > 2), Michelet *et al.* found that the GCS score and the Head AIS were associated with pneumonia occurring within the first 72 hours [16]. The Head AIS was also an independent risk factor associated with pneumonia in another study assessing risk factors for pneumonia in 571 severely trauma patients receiving initial invasive

**Table 2. Multivariate analysis for prediction of VAP occurrence.**

| Model 1 (n = 688) | OR | Inf CI | Sup CI | p-value | VIF |
|---|---|---|---|---|---|
| Age (years) | 1.00 | 0.99 | 1.01 | 0.941 | 1.35 |
| Male gender | 2.77 | 1.68 | 4.77 | **< 0.001** | 1.03 |
| ASA score > 1 | 1.64 | 1.08 | 2.50 | **0.019** | 1.32 |
| AIS Abdomen > 2 | 1.55 | 0.99 | 2.39 | 0.053 | 1.09 |
| Absolute thoracic AIS value | 1.51 | 1.14 | 1.99 | **0.003** | 1.10 |
| Prehospital antibiotic administration | 0.54 | 0.29 | 0.94 | **0.028** | 1.02 |
| Spinal cord lesion with neurologic deficit | 1.51 | 0.70 | 3.12 | 0.283 | 1.09 |
| Massive transfusion | 1.80 | 0.90 | 3.55 | 0.096 | 1.09 |
| Initial GCS score < 9 | 2.71 | 1.88 | 3.96 | **< 0.001** | 1.04 |
| Model 2 (n = 688) | | | | | |
| Age (years) | 1.00 | 0.99 | 1.01 | 0.949 | 1.38 |
| Male gender | 2.72 | 1.65 | 4.66 | **< 0.001** | 1.03 |
| ASA > 1 | 1.71 | 1.13 | 2.59 | **0.011** | 1.30 |
| ISS > 15 | 3.15 | 1.13 | 11.99 | **0.025** | 1.02 |
| Prehospital antibiotic administration | 0.51 | 0.27 | 0.90 | **0.018** | 1.03 |
| Lung contusion | 1.23 | 0.82 | 1.84 | 0.321 | 1.08 |
| Flail chest | 1.24 | 0.70 | 2.15 | 0.451 | 1.06 |
| Hemothorax | 1.40 | 0.93 | 2.10 | 0.105 | 1.07 |
| Spinal cord lesion with neurologic deficit | 1.81 | 0.87 | 3.66 | 0.112 | 1.04 |
| Number of transfused PRBCs in the first 24 h | 1.04 | 1.00 | 1.08 | **0.027** | 1.11 |
| Initial GCS score < 9 | 2.57 | 1.79 | 3.74 | **< 0.001** | 1.03 |

ASA: American society of anesthesiology; AIS: Abbreviated injury score; GCS: Glasgow coma scale; ISS: injury severity score; VAP: Ventilator associated pneumonia; VIF: Variance Inflation Factor PRBCs: packed red blood cells

mechanical ventilation [15]. The largest study describing risk factors for pneumonia in traumatic brain injury patients receiving invasive mechanical ventilation was performed by Robba et al.[3]. This retrospective multicenter cohort study from the CENTER-TBI study data included 962 patients with traumatic brain injury receiving invasive mechanical ventilation for more than 48 hours and staying in the ICU for more than 72 hours [3]. A GCS score < 9 and the motor component of GCS score were associated with VAP in univariate analyses, although the multivariate analysis did not confirm this finding [3]. In this cohort, the presence of associated chest trauma was as an independent risk factor for VAP with a hazard ratio of 1.40 [CI95%: 1.03–1.90] [3]. Thus, severe thoracic associated with severe traumatic brain injury was associated with high rates of pneumonia.

The results of our study did not confirm massive transfusion as a risk factor associated with suspected VAP (OR = 1.80 [CI95%: 0.90; 3.55]). However, the number of PRBCs was associated with early suspected VAP in one of the multivariate analysis models (OR = 1.04 [CI95%: 1.00–1.08]; p = 0.03). In our cohort, the lack of association between massive transfusion and the occurrence of early suspected VAP could be explained by the exclusion of patients not surviving within the first five days who were the most susceptible to be massively transfused. Furthermore the study was not designed for this specific end-point and may lack power. Nevertheless, blood product transfusion was frequently described as being associated with an increased risk of pneumonia. Bochicchio et al., in a cohort of 766 ICU trauma patients receiving more than 48 hours of invasive mechanical ventilation, identified blood product transfusion as an independent risk factor associated with pneumonia, regardless of the type of blood product transfused [17]. Torrance et al., in a study including 112 ICU trauma patients mechanically ventilated, measured a set of biomarkers including interleukin-10 (IL-10), Foxp3, TNF

alpha, and interferon gamma [18]. They found an association between blood transfusion within the first 24 h and increased serum levels of IL-10 reflecting an immunosuppressive response, which could explain the increased risk for VAP due to an immunosuppression associated with blood transfusion [18].

Male gender and ASA score were associated with more suspected VAP in our cohort such as severity of thoracic injury and general severity of the trauma. These findings are in line with previous studies on the topic. In a prospective study Michet *et al.* described an association between thoracic AIS and increase risk of pneumonia [16]. Indeed, on two larges epidemiologic cohorts male gender, preexisting disease where associated with an increasing risk of sepsis [19,20]. Severity of trauma was also associated with a six-fold to 16-fold increase incidence of sepsis for moderate (ISS 15–29) and severe injury (ISS > 29) compared with mild injury defined as an ISS < 15 [19].

Finally, we found that prehospital antibiotic therapy was a protective factor associated with a decreased rate of early suspected VAP. This finding confirmed those reported in previous studies [21]. In guidelines, use of prehospital antibiotic therapy in justified by open bone fractures infection prevention [22]. This practice seems to be associated with less early suspected VAP occurrence for patients with risk factors. In trauma patients with severe traumatic brain injury receiving invasive mechanical ventilation, Robba *et al.* highlighted that the administration of antibiotic prophylaxis was a protective factor associated with a decreased risk for pneumonia (hazard ratio (HR) of 0.69 [95%CI: 0.50–0.96]) [3]. Recently Dahyot-Fizelier *et al.,* in a randomized clinical trial including 345 patients with severe traumatic brain injury requiring mechanical ventilation, showed that one single dose of ceftriaxone reduced the occurrence of early VAP (HR 0·60 [95% CI 0·38–0·95], p = 0·030) [23]. This result offer opportunities for further studies on the prevention or early VAP in the context of trauma.

Our study has some limitations. First, the diagnosis of VAP is declarative, which could introduce a level of variability between centers. However, our data were collected from French ICUs in which the diagnosis of VAP was based on our national guidelines and reflected a pragmatic approach [8,10]. Nevertheless we qualified the diagnosis of VAP as suspected because no microbiological documentation was reported in the data base. The choice of our first outcome may be matter of debate. In our study, we focused on early suspected VAP (occurring from 48 h to day 5 after ICU admission). This choice reduced the number of patients but avoided a potential confounding bias in the early phase due to premature deaths. Indeed, pneumonia remains a difficult diagnosis and therefore could be misdiagnosed in patients with pulmonary contusions or in inhalation context at the initial phase, i.e., in the first 48 h. Finally, we excluded a large number of patients due to missing data, which is inherent to the study design. Despite these exclusions, our study represents, to our knowledge, the largest cohort of severely blunt chest trauma patients admitted to ICU and receiving invasive mechanical ventilation. Finally, logistic regression identified pre-hospital antibiotics therapy as a protective factor, this result is in line with recent finding as describe above but the database was not able to provide information about the duration of the antibiotic therapy after admission. This bias may affect the interpretation of the finding.

## Conclusion

In conclusion, in patients with severe blunt chest trauma patients receiving invasive mechanical ventilation, male gender, ASA score >1, ISS > 15, initial GCS score < 9, absolute thoracic AIS value, and the number of PRBCs transfused were independent risk factors associated with early suspected VAP. Administration of pre-hospital antibiotic therapy could be an independent protective factor for early suspected VAP. This point may lead to future studies in VAP prevention in the trauma context.

## Supporting information

**S1 File.  STROBE Statement—Checklist of items that should be included in reports of case-control studies.** (DOC)

**S1 Table. Descriptive statistics and univariate analysis comparing the suspected VAP and non-suspected VAP group. Quantitative variables are expressed in medians and interquartile ranges (25-75) or in means ± standard deviations depending on their distribution.**
(DOC)

## Author contributions

**Conceptualization:** Gary Duclos.

**Data curation:** Gary Duclos, Victor Vidal, Caroline Jeantrelle, Véronique Ramonda, Paer-Selim Abback.

**Formal analysis:** Gary Duclos, Noemie Resseguier.

**Investigation:** Gary Duclos.

**Methodology:** Gary Duclos, Noemie Resseguier, Paer-Selim Abback.

**Project administration:** Gary Duclos, Caroline Jeantrelle, Véronique Ramonda, Paer-Selim Abback, Julien Pottecher, Marc Leone.

**Resources:** Anh Dao Phan, Jean-Luc Hanouz, Véronique Ramonda, Julien Pottecher.

**Software:** Gary Duclos.

**Supervision:** Marc Leone.

**Validation:** Mathieu Boutonnet, Thomas Botrel, Gerard Audibert, Manon Seguret, Vincent Legros, Caroline Jeantrelle, Anh Dao Phan, Olivier Langeron, Elisabeth Gaertner, Jean-Luc Hanouz, Thomas Clavier, Fanny Bounes, Paer-Selim Abback, Mathieu Willig, Julien Pottecher, Marc Leone.

**Visualization:** Véronique Ramonda, Marc Leone.

**Writing – original draft:** Gary Duclos, Marc Leone.

**Writing – review & editing:** Mathieu Boutonnet, Thomas Botrel, Gerard Audibert, Manon Seguret, Vincent Legros, Caroline Jeantrelle, Anh Dao Phan, Olivier Langeron, Elisabeth Gaertner, Jean-Luc Hanouz, Thomas Clavier, Fanny Bounes, Véronique Ramonda, Paer-Selim Abback, Mathieu Willig, Julien Pottecher, Marc Leone.

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
