## [Decision Letter · Decision Letter 0]

14 Jan 2025

Dear Dr. Duclos,

Thank you for submitting your manuscript to PLOS ONE. After careful consideration, we feel that it has merit but does not fully meet PLOS ONE’s publication criteria as it currently stands. Therefore, we invite you to submit a revised version of the manuscript that addresses the points raised during the review process.

We look forward to receiving your revised manuscript.

Kind regards,

Jean Baptiste Lascarrou

Academic Editor

PLOS ONE

**Journal Requirements:**

2. In the online submission form, you indicated that your data is available only on request from a third party. Please note that your Data Availability Statement is currently missing the contact details for the third party, such as an email address or a link to where data requests can be made. Please update your statement with the missing information. 

5. One of the noted authors is a group or consortium "TraumaBase Group". In addition to naming the author group, please list the individual authors and affiliations within this group in the acknowledgments section of your manuscript. Please also indicate clearly a lead author for this group along with a contact email address.

Reviewers' comments:

Reviewer's Responses to Questions

**Comments to the Author**

1. Is the manuscript technically sound, and do the data support the conclusions?

Reviewer #1: Yes

Reviewer #2: Yes

Reviewer #3: Yes

2. Has the statistical analysis been performed appropriately and rigorously?

Reviewer #1: Yes

Reviewer #2: No

Reviewer #3: Yes

3. Have the authors made all data underlying the findings in their manuscript fully available?

Reviewer #1: Yes

Reviewer #2: Yes

Reviewer #3: Yes

4. Is the manuscript presented in an intelligible fashion and written in standard English?

Reviewer #1: Yes

Reviewer #2: Yes

Reviewer #3: Yes

**Reviewer #1: ** Thank you for your work, which explores a little more the Traumabase in a little-studied area of blunt thoracic trauma reveiving mechanical ventilation.

You rightly note that your 30-days mortality is low because you exluded the most severe patients inducing a non-exposure bias and the possible lack of association with massive transfusion.

Concerning the statistical analysis : Why have you runned two multivariate models that include redundancies ? If you use two different models, you may be tempted to choose the model that performs better. You also conclude with the results of the two models but I don't see how each model offers anything different.

I would have liked to know a little more about the micro-organisms found (and a possible difference with pure traumatic brain injuries). I found this work very interesting, it prompts reflection on VAP prevention. The role of prehospital antibiotic therapy and its generalisation beyond open bones fractures infections prevention is a field of research.

**Reviewer #2:**  I thank the editor for the opportunity to evaluate this manuscript, which aims to identify risk factors for early VAP in patients admitted for severe thoracic blunt trauma and requiring mechanical ventilation in France between 2014 and 2021.

The strengths of this work lie in:

• A large, homogeneous national cohort (sourced from the Traumabase registry),

• Clearly stated objectives and methodology,

• A manuscript that is overall well-written.

Below are my comments:

Major comments:

1. Definition of VAP:

The definition of VAP used in this article is insufficiently detailed. The declarative nature of the diagnosis and the absence of an adjudication committee represent significant limitations, especially in a population where, as the authors rightly highlight, differential diagnoses are common. While the authors explain the criteria for suspecting VAP, they do not mention any qualitative or quantitative microbiological criterion. It appears that the mere initiation of empirical antibiotic therapy for suspected VAP was sufficient for diagnosis. Is the rate of microbiologically documented VAP known in this cohort? If no microbiological criterion was required, I suggest replacing the term "VAP" with "suspected VAP" throughout the manuscript.

2. Statistical methodology:

The authors used a logistic regression model for their analysis. To account for bias related to mechanical ventilation exposure, they excluded patients who died before day 5. However, they did not exclude patients extubated before day 5, which raises questions. This methodology does not appear optimal to address the notable competing risks of death and extubation. A Fine and Gray model would be more appropriate, as it accounts for the time to VAP occurrence (like a Cox model) and the cumulative incidence of events (like a logistic model).

3. Flow Chart details:

The flow chart mentions exclusion criteria that are not detailed in the main text, such as pneumonia identified before 48 hours of mechanical ventilation (a reasonable criterion) and late-onset pneumonia, which is unexpected.

4. Prehospital antibiotics as a protective factor:

Logistic regression identified prehospital antibiotic administration as a protective factor against VAP. This is an interesting finding, echoing recent literature advocating very short-duration antibioprophylaxis (1 dose to 3 days) to prevent VAP in high-risk ICU patients. However, in this cohort, patients received prehospital antibiotics primarily for associated open fractures. The authors do not specify whether antibiotics were continued after hospital admission and their duration, which significantly affects the interpretation and message of this result.

5. Discussion Section:

Some factors identified as independently associated with VAP diagnosis are not discussed, such as male gender, ASA score, ISS, and absolute thoracic AIS value. The authors' explanation for the non-significant association between massive transfusion and VAP could be clarified by simply noting that the number of events was likely insufficient, and the study was underpowered to detect this association.

Minor comments:

1. Paragraph numbering is inconsistent.

2. Author affiliations are repetitive.

3. The citation formatting needs revision (references should be placed before the period, not after).

4. The “Definitions of Analyzed Variables” section contains redundancies and could be streamlined.

5. The variables included in the two logistic regression models differ between Table 2 and the text (e.g., ASA score is missing in the text for the second model).

6. Use "our study" instead of "this study" in the discussion section to improve clarity.

7. Page 14, lines 274-275: The odds ratio mentioned in the text does not match the results in Table 2.

8. The conclusion omits ASA score from the list of factors independently associated with VAP.

9. Several English language errors require correction, such as:

o "our study was aimed,"

o "who were the most susceptible to received massive transfusion,"

o "biomarkers of IL-10,"

o "are reflection a pragmatic approach,"

o "was could be an independent protective factor."

10. The abbreviation AIS is defined differently in the abstract and main text ("acute injury score" vs. "abbreviated injury scale").

11. The ISS abbreviation is not introduced in the main text.

12. Using “ISS score” or “AIS score” is redundant.

13. Replace "male sex" with "male gender."

Thank you to the authors for their work on this important topic. I hope these comments will be helpful in refining and strengthening this valuable contribution to the field.

**Reviewer #3: ** Thank you to the authors for the opportunity to review their well written manuscript. I applaud their efforts to further define a distinct population of patients with blunt thoracic injury as separate from a general ICU population at risk for VAP. I have a few questions for the authors:

1. The authors describe the selection of variables for the multivariable models as “a priori from the literature.” Could the authors provide more detail about how these variables were selected, and which studies influenced the decisions?

2. Was a larger model run that included variables from both smaller models? The exclusion of certain variables clouds the conclusions drawn when, for example ISS, is excluded in one of the models.

3. Did the authors find particular types of thoracic injuries to be more significant in any models run? In Model 2, lung contusion, flail chest, and hemothorax are all included together which may represent significant collinearity.

4. In the methods section the authors note that they included patients who received mechanical ventilation for “any length of time.” Were patients intubated for the entire 5-day period of interest or were some patients extubated in this period, but still found to have VAP?

5. In both multivariable models variables such as ASA >1, AIS >2, ISS >15 were included. Were these categorical variables? How were these cutoffs determined? An ASA patient of 4 is quite different from an ASA 2, but these are treated the same in both models.

6. The authors mention in the discussion that GCS <9 was used as a substitute for aspiration event. While I agree that aspiration is difficult to assess clinically, could the authors further comment on this decision? Many patients with aspiration do not have a traumatic brain injury, and vice versa.

7. The increased risk of the combination of TBI and thoracic injury is very interesting. Were there enough patients in this group to perform any subgroup analyses?

8. Did any patients undergo surgical fixation of rib fractures and did the affect the rates of VAP?

9. Given the identification of particular risk factors, do the authors suggest any change in practice to help prevent VAP in the higher risk patients?

Overall, this is an excellent study that will expand the literature on this important topic. Trauma patients, and particularly those with blunt thoracic trauma, are certainly high risk for pulmonary infection and should be considered distinct from a general ICU patient population. Thank you again to the authors for this opportunity.

**Do you want your identity to be public for this peer review?** For information about this choice, including consent withdrawal, please see our Privacy Policy

Reviewer #1: **Yes: ** Alejandro Villaamil

Reviewer #2: No

Reviewer #3: **Yes: ** Kevin N Harrell

---

## [Author Response · Author response to Decision Letter 0]

27 Mar 2025

Reviewer #1: Thank you for your work, which explores a little more the Traumabase in a little-studied area of blunt thoracic trauma reveiving mechanical ventilation.

You rightly note that your 30-days mortality is low because you exluded the most severe patients inducing a non-exposure bias and the possible lack of association with massive transfusion.

Concerning the statistical analysis : Why have you runned two multivariate models that include redundancies ? If you use two different models, you may be tempted to choose the model that performs better. You also conclude with the results of the two models but I don't see how each model offers anything different.

Dear reviewer, thanks for the time you ve spent in reading this manuscript.

We wanted to explore 2 different kind of injury what was susceptible to lead to pneumonia.

First we wanted to evaluate the global severity of thoracic injury by performing the first model and then we wanted to explore some specific kind of lesions as contusion, hemothoraces or chest flail. We were not able to put all the variable in a same model because of redundancies so we ve chosen to run 2 models.

To rule out any redundancies in the model we have now added the Variance inflation factors which remains low for every variable.

I would have liked to know a little more about the micro-organisms found (and a possible difference with pure traumatic brain injuries). I found this work very interesting, it prompts reflection on VAP prevention. The role of prehospital antibiotic therapy and its generalisation beyond open bones fractures infections prevention is a field of research.

Unfortunately the microbiological characteristics of the VAP were not available in the data base. This is, indeed a major bias of the study. Even when following guidelines, pneumonia remains a difficult diagnosis. We have tried to attenuate it by exploring only pneumonia occurring after 2 days of admission in ICU. We supposed that physicians would be less susceptible to over treat simple pneumonia suspicion or confounding diagnosis such as contusions.

Reviewer #2: I thank the editor for the opportunity to evaluate this manuscript, which aims to identify risk factors for early VAP in patients admitted for severe thoracic blunt trauma and requiring mechanical ventilation in France between 2014 and 2021.

The strengths of this work lie in:

• A large, homogeneous national cohort (sourced from the Traumabase registry),

• Clearly stated objectives and methodology,

• A manuscript that is overall well-written.

Below are my comments:

Major comments:

1. Definition of VAP:

The definition of VAP used in this article is insufficiently detailed. The declarative nature of the diagnosis and the absence of an adjudication committee represent significant limitations, especially in a population where, as the authors rightly highlight, differential diagnoses are common. While the authors explain the criteria for suspecting VAP, they do not mention any qualitative or quantitative microbiological criterion. It appears that the mere initiation of empirical antibiotic therapy for suspected VAP was sufficient for diagnosis. Is the rate of microbiologically documented VAP known in this cohort? If no microbiological criterion was required, I suggest replacing the term "VAP" with "suspected VAP" throughout the manuscript.

Dear reviewer, thank you very much for the time you ve spent in correcting and improving this manuscript. Our remarks will be very useful.

Regarding the diagnosis of VAP, as you ve highlighted, its declarative nature is a major bias and no microbiological document was available to prevent it. As you suggested we have replaced the VAP term with suspected VAP when it was referring our data.

2. Statistical methodology:

The authors used a logistic regression model for their analysis. To account for bias related to mechanical ventilation exposure, they excluded patients who died before day 5. However, they did not exclude patients extubated before day 5, which raises questions. This methodology does not appear optimal to address the notable competing risks of death and extubation. A Fine and Gray model would be more appropriate, as it accounts for the time to VAP occurrence (like a Cox model) and the cumulative incidence of events (like a logistic model).

Dear reviewer, thank you for this remark.

To simplify the understanding of the work and to clarify the take home message for the readers we have chosen to remove all patients without at least 5 days of mechanical ventilation from the analysis. 5 patients were excluded and the analysis and tables have been totally remade.

New results stay coherent with the previous one but we can now insure the lack of exposition bias regarding the exposure of mechanical ventilation.

3. Flow Chart details:

The flow chart mentions exclusion criteria that are not detailed in the main text, such as pneumonia identified before 48 hours of mechanical ventilation (a reasonable criterion) and late-onset pneumonia, which is unexpected.

Dear reviewer, the flow chart and the main text are now coherent. The misunderstanding provides from the inclusion criterion of early onset suspected VAP (between 2 and 5 days of mechanical ventilation exposure) that forces the exclusion of immediate pneumonia and late VAP.

Sorry for this mistake.

4. Prehospital antibiotics as a protective factor:

Logistic regression identified prehospital antibiotic administration as a protective factor against VAP. This is an interesting finding, echoing recent literature advocating very short-duration antibioprophylaxis (1 dose to 3 days) to prevent VAP in high-risk ICU patients. However, in this cohort, patients received prehospital antibiotics primarily for associated open fractures. The authors do not specify whether antibiotics were continued after hospital admission and their duration, which significantly affects the interpretation and message of this result.

Dear reviewer, you re right but unfortunately this information is not available in the data base. We have added this limitation in the discussion.

5. Discussion Section:

Some factors identified as independently associated with VAP diagnosis are not discussed, such as male gender, ASA score, ISS, and absolute thoracic AIS value. The authors' explanation for the non-significant association between massive transfusion and VAP could be clarified by simply noting that the number of events was likely insufficient, and the study was underpowered to detect this association.

Dear reviewer, I have added some discussion point about the points you have highlighted.

Minor comments:

1. Paragraph numbering is inconsistent.

Dear reviewer, we ve corrected this point. Thank you.

2. Author affiliations are repetitive.

3. The citation formatting needs revision (references should be placed before the period, not after).

This point has been corrected. Thank you.

4. The “Definitions of Analyzed Variables” section contains redundancies and could be streamlined.

Dear reviewer, this section has been slightly reworked.

5. The variables included in the two logistic regression models differ between Table 2 and the text (e.g., ASA score is missing in the text for the second model).

Dear reviewer, thank you. This has been corrected.

6. Use "our study" instead of "this study" in the discussion section to improve clarity.

Dear reviewer, this has been corrected.

7. Page 14, lines 274-275: The odds ratio mentioned in the text does not match the results in Table 2.

Dear reviewer, this point has been corrected. Thank you.

8. The conclusion omits ASA score from the list of factors independently associated with VAP.

This point has been corrected.

9. Several English language errors require correction, such as:

o "our study was aimed,"

o "who were the most susceptible to received massive transfusion,"

o "biomarkers of IL-10,"

o "are reflection a pragmatic approach,"

o "was could be an independent protective factor."

10. The abbreviation AIS is defined differently in the abstract and main text ("acute injury score" vs. "abbreviated injury scale").

11. The ISS abbreviation is not introduced in the main text.

12. Using “ISS score” or “AIS score” is redundant.

13. Replace "male sex" with "male gender."

Dear reviewer, all errors have been corrected. Thank you for your helpful reviewing

Thank you to the authors for their work on this important topic. I hope these comments will be helpful in refining and strengthening this valuable contribution to the field.

Reviewer #3: Thank you to the authors for the opportunity to review their well written manuscript. I applaud their efforts to further define a distinct population of patients with blunt thoracic injury as separate from a general ICU population at risk for VAP. I have a few questions for the authors:

1. The authors describe the selection of variables for the multivariable models as “a priori from the literature.” Could the authors provide more detail about how these variables were selected, and which studies influenced the decisions?

Dear reviewer, thank you for the time you ve spent in reading and correcting this work.

We have added the reference inside the material and method section. Also, variables are extensively discussed in the appropriate section.

2. Was a larger model run that included variables from both smaller models? The exclusion of certain variables clouds the conclusions drawn when, for example ISS, is excluded in one of the models.

Dear reviewer, thank you for this good question. Indeed we had issues in having a larger model for some reason. First, the number of variable is limited depending the number of events. In this case, we could have added some more variable but the variance interference factors were raising. That could have introduced some redundancies into the model. So we have chosen 2 different approaches, the first was to focus about the global severity of the whole trauma (ISS) and adding some specific kind of lesion. The second was to focus about the severity of the thoracic injuries (AIS thoracic).

3. Did the authors find particular types of thoracic injuries to be more significant in any models run? In Model 2, lung contusion, flail chest, and hemothorax are all included together which may represent significant collinearity.

Dear reviewer, we have checked the collinearity of the variable by calculating the VIF. We have added them in the table 2. All VIF were low in both model, insuring the no-collinearity of them.

4. In the methods section the authors note that they included patients who received mechanical ventilation for “any length of time.” Were patients intubated for the entire 5-day period of interest or were some patients extubated in this period, but still found to have VAP?

Dear Reviewer, this is an excellent question.

Indeed after checking the data we found that 5 patients were weaned of mechanical ventilation before day 5. We wanted to present clean results so excluded the patients and made the whole analysis again. That explains why we took so long before responding the reviewing. Guaranteed without exposition bias this time.

5. In both multivariable models variables such as ASA >1, AIS >2, ISS >15 were included. Were these categorical variables? How were these cutoffs determined? An ASA patient of 4 is quite different from an ASA 2, but these are treated the same in both models.

Dear reviewer, we wanted to class comorbid and no-comorbid patient by this bias. As you can see in table 1, the number of comorbid patients was very low (but populations of trauma patients are often young). The number of patients presenting ASA 3 and 4 was too low to be analyzed in an efficient way. This is why we have chosen to determine the ASA cut-off of 2.

6. The authors mention in the discussion that GCS <9 was used as a substitute for aspiration event. While I agree that aspiration is difficult to assess clinically, could the authors further comment on this decision? Many patients with aspiration do not have a traumatic brain injury, and vice versa.

Dear reviewer, this decision is indeed arguable. We wanted to prevent any bias and redundancies for the logistic models. The most severe risk for aspiration event seems to be coma whatever its cause. By choosing a GCS < 9 we think that we included most situations with risk of aspiration events (TBI, cardiac arrest, intoxication).

7. The increased risk of the combination of TBI and thoracic injury is very interesting. Were there enough patients in this group to perform any subgroup analyses?

Dear reviewer, unfortunately the number of patients was too low to perform a subgroup analyses. Moreover we wanted to keep the manuscript clear and adding another subgroup could have complicated the analysis and the understanding of the text for an underpowered side analysis.

8. Did any patients undergo surgical fixation of rib fractures and did the affect the rates of VAP?

Dear reviewer, 75 rib flail chest were reported in our cohort but only 8 patients underwent a rib fracture surgery. It was not enough to be analyzed apart. (Data are available in supplemental table 1).

9. Given the identification of particular risk factors, do the authors suggest any change in practice to help prevent VAP in the higher risk patients?

Dear reviewer, thank you for this question. It is probability to soon considering the data available to recommend extensive use of short duration antibiotic use at trauma admission. Nevertheless we are working on an RCT at the moment to explore this hypothesis

Overall, this is an excellent study that will expand the literature on this important topic. Trauma patients, and particularly those with blunt thoracic trauma, are certainly high risk for pulmonary infection and should be considered distinct from a general ICU patient population. Thank you again to the authors for this opportunity.

---

## [Decision Letter · Decision Letter 1]

17 Apr 2025

Dear Dr. Duclos,

Thank you for submitting your manuscript to PLOS ONE. After careful consideration, we feel that it has merit but does not fully meet PLOS ONE’s publication criteria as it currently stands. Therefore, we invite you to submit a revised version of the manuscript that addresses the points raised during the review process.

We look forward to receiving your revised manuscript.

Kind regards,

Jean Baptiste Lascarrou

Academic Editor

PLOS ONE

Journal Requirements:

Reviewers' comments:

Reviewer's Responses to Questions

**Comments to the Author**

Reviewer #1: All comments have been addressed

Reviewer #2: (No Response)

Reviewer #3: All comments have been addressed

2. Is the manuscript technically sound, and do the data support the conclusions?

Reviewer #1: Yes

Reviewer #2: Yes

Reviewer #3: Partly

3. Has the statistical analysis been performed appropriately and rigorously?

Reviewer #1: Yes

Reviewer #2: Yes

Reviewer #3: Yes

4. Have the authors made all data underlying the findings in their manuscript fully available?

Reviewer #1: Yes

Reviewer #2: Yes

Reviewer #3: Yes

5. Is the manuscript presented in an intelligible fashion and written in standard English?

Reviewer #1: Yes

Reviewer #2: Yes

Reviewer #3: Yes

Reviewer #1: Thank you for your corrections. As you say in the discussion, the declarative nature of the diagnosis, the absence of an adjudication committee and the absence of any microbiological features represent significant limitations.

Reviewer #2: I would like to thank the authors for addressing my comments and revising the manuscript accordingly. I appreciate the effort made to improve the clarity and precision of the text.

However, I noticed that the term "suspected VAP" has been incorrectly replaced with "suggested VAP" throughout the manuscript. The term "suggested VAP" is not appropriate and should be corrected to "suspected VAP" everywhere. Additionally, the title should clearly indicate that the study focuses on early suspected VAP.

Some other points remain unaddressed:

• The conclusion of the abstract still omits ASA score from the list of factors independently associated with VAP.

• The abbreviation AIS is still inconsistently defined in the abstract and main text ("acute injury score" vs. "abbreviated injury scale").

• The redundancy in "ISS score" has not been fully corrected, as "ISS score" still appears in some instances.

• "Male sex" should be replaced with "male gender" throughout the manuscript.

Finally, in the second-to-last sentence before the conclusion on page 15, "this results" should be corrected to "this result."

I have no further comments.

I wish the authors much success in their future interventional project, which may further validate the preliminary findings of this study. It’s great to see such work progressing in this field!

Reviewer #3: Thank you for your response to my comments. The authors have addressed all my concerns and strengthened this paper.

**Do you want your identity to be public for this peer review?** For information about this choice, including consent withdrawal, please see our Privacy Policy

Reviewer #1: **Yes: ** Alejandro Villaamil

Reviewer #2: No

Reviewer #3: **Yes: ** Kevin Harrell, MD

---

## [Author Response · Author response to Decision Letter 1]

18 Apr 2025

6. Review Comments to the Author

Reviewer #1: Thank you for your corrections. As you say in the discussion, the declarative nature of the diagnosis, the absence of an adjudication committee and the absence of any microbiological features represent significant limitations.

Reviewer #2: I would like to thank the authors for addressing my comments and revising the manuscript accordingly. I appreciate the effort made to improve the clarity and precision of the text.

However, I noticed that the term "suspected VAP" has been incorrectly replaced with "suggested VAP" throughout the manuscript. The term "suggested VAP" is not appropriate and should be corrected to "suspected VAP" everywhere. Additionally, the title should clearly indicate that the study focuses on early suspected VAP.

Dear reviewer, this mistake has been corrected everywhere in the text and tables

Some other points remain unaddressed:

• The conclusion of the abstract still omits ASA score from the list of factors independently associated with VAP.

Dear reviewer, ASA score has been added

• The abbreviation AIS is still inconsistently defined in the abstract and main text ("acute injury score" vs. "abbreviated injury scale").

Dear Reviewer, sorry for this omission. It has been corrected

• The redundancy in "ISS score" has not been fully corrected, as "ISS score" still appears in some instances.

Dear reviewer, this redundancy has been corrected.

• "Male sex" should be replaced with "male gender" throughout the manuscript.

Dear Reviewer, it has been corrected

Finally, in the second-to-last sentence before the conclusion on page 15, "this results" should be corrected to "this result."

Dear Reviewer, it has been corrected

I have no further comments.

I wish the authors much success in their future interventional project, which may further validate the preliminary findings of this study. It’s great to see such work progressing in this field!

Reviewer #3: Thank you for your response to my comments. The authors have addressed all my concerns and strengthened this paper.

---

## [Editor Report · Decision Letter 2]

22 Apr 2025

Risk factors for early suspected ventilator-associated pneumonia in severe thoracic blunt trauma patient: a French national cohort study.

PONE-D-24-45522R2

Dear Dr. Duclos,

We’re pleased to inform you that your manuscript has been judged scientifically suitable for publication and will be formally accepted for publication once it meets all outstanding technical requirements.

Kind regards,

Jean Baptiste Lascarrou

Academic Editor

PLOS ONE
---

## [Editor Report · Acceptance letter]

PONE-D-24-45522R2

PLOS ONE

Dear Dr. Duclos,

I'm pleased to inform you that your manuscript has been deemed suitable for publication in PLOS ONE. Congratulations! Your manuscript is now being handed over to our production team.

Kind regards,

on behalf of

Dr. Jean Baptiste Lascarrou

Academic Editor

PLOS ONE